# Clinical characteristics and risk factors for rebleeding in uremic patients with obscure gastrointestinal bleeding undergoing deep enteroscopy: A multi-center retrospective study in Taiwan

Hsuan-Jen Hung[1], Chen-Shuan Chung[2,3,4], Chi-Ming Tai[4,5], Chen-Wang Chang[4,6,7], Chao-Ming Tseng[5], Go-Shine Huang[8], Meng-Chiung Lin[9], Tien-Yu Huang[1,4]*

1 Division of Gastroenterology, Department of Internal Medicine, Tri-Service General Hospital, National Defense Medical Center, Taipei, Taiwan, 2 Division of Gastroenterology and Hepatology, Department of Internal Medicine, Far Eastern Memorial Hospital, New Taipei City, Taiwan, 3 College of Medicine, Fu Jen Catholic University, New Taipei City, Taiwan, 4 Taiwan Association for the Study of Small Intestinal Diseases, Taoyuan, Taiwan, 5 Division of Gastroenterology and Hepatology, Department of Internal Medicine, E-Da Hospital, I-Shou University, Kaohsiung City, Taiwan, 6 Division of Gastroenterology, Department of Internal Medicine, Mackay Memorial Hospital, Taipei, Taiwan, 7 MacKay Junior College of Medicine, Nursing and Management, Taipei, Taiwan, 8 Department of Anesthesiology, Tri-Service General Hospital, National Defense Medical Center, Taipei, Taiwan, 9 Division of Gastroenterology, Department of Internal Medicine, Taichiung Armed Forces General Hospital, Taichiung, Taiwan

* tienyu27@gmail.com

## Abstract

### Background/Purpose

Recurrent obscure gastrointestinal bleeding (OGIB) in patients with chronic kidney disease is a challenge often faced by physicians, given the need for repeated hospitalizations, multiple extensive examinations, limited treatment options, and high medical costs. The purpose of this study was to identify the clinical characteristics of uremic patients undergoing deep enteroscopy for OGIB and analyze the risk factors for rebleeding in these patients after undergoing single-balloon enteroscopy (SBE).

### Methods

Out of 765 patients with OGIB who underwent 1004 procedures of SBE in four teaching hospitals, 78 uremic patients with OGIB were enrolled. Clinical characteristics and endoscopic findings were collected, and multiple variables were analyzed to determine the risk of rebleeding after SBE.

### Results

The diagnostic yield was 75.6%, and the rebleeding rate was 29.5% in the enrolled uremic patients. The most common etiology was angiodysplasia (74.6%) and the most common site was the jejunum (50.8%). The endoscopic intervention rate was 62.8% and most patients were treated with argon plasma coagulation (75.6%). Among the eight patients with

**Data Availability Statement:** All relevant data are within the manuscript and its Supporting information files.

**Funding:** The research funding for this study was supported by the Department of Defense (MND-MAB-D-111114) and Tri-Service General Hospital (TSGH-D-109073), Taiwan. The funders had no role in study design, data collection and analysis, decision to publish, or preparation of the manuscript.

**Competing interests:** The authors have declared that no competing interests exist.

valvular heart disease (VHD), four (50%) had severe aortic stenosis, and the remaining had non-aortic stenosis-VHD. VHD (p < 0.05) and angiodysplasia (p < 0.05) were both associated with a higher rebleeding rate.

## Conclusion

VHD may be an independent risk factor associated with rebleeding after SBE in uremic patients with OGIB. Moreover, uremic patients with angiodysplasia-related bleeding appear to have a higher rebleeding rate than those with alternative causes of bleeding.

## Introduction

Obscure gastrointestinal bleeding (OGIB) is defined as persistent or recurrent bleeding of unknown origin after negative endoscopic results (esophagogastroduodenoscopy (EGD) and colonoscopy) [1]. Older patients undergoing dialysis are particularly prone to OGIB, and vascular lesions (e.g., angiodysplasia) of the small intestine are considered to be the most common cause of OGIB in this patient group [2]. Angiodysplasia has been reported in elderly populations and patients with certain predisposing conditions, such as end-stage renal disease, liver disease, aortic stenosis, and Von Willebrand disease [3, 4]. Angiodysplasia is defined as a vascular malformation consisting of dilated and curved arterial or venous capillaries, usually <5 mm in diameter, located in the mucosa and submucosa of the gastrointestinal tract [5]. Although angiodysplasia can occur in any part of the digestive tract, it is usually the cause of repeated bleeding in the small intestine and is more difficult to identify and manage [2].

In patients with chronic kidney disease (CKD), a higher incidence of angiodysplasia of the stomach or proximal small intestine has been reported. Most patients have no symptoms, but some can experience gastrointestinal bleeding. If the bleeding lesion is in the stomach, duodenum, colon, or terminal ileum, it can be treated with a traditional EGD or colonoscopy. In contrast, if the lesion is located in the small intestine, it is challenging to approach the site of the bleeding or manage the bleeding further. However, there are several advanced tools for the management of OGIB; in addition to traditional methods, capsule endoscopy and device-assisted enteroscopy (e.g., single-balloon or double-balloon enteroscopy) could be useful in the diagnosis and treatment of small intestinal diseases [6].

Single-balloon enteroscopy (SBE) is a useful endoscopic tool for both the diagnosis and treatment of OGIB. The diagnosis rate of SBE is around 58–70% in patients with OGIB [7, 8]. In our previous study, the diagnostic yield of SBE was 61%, and the endoscopic intervention rate was 20.5%. In addition, chronic renal failure is a common risk factor for OGIB and OGIB with rebleeding. Another study reported that small bowel angiodysplasia was identified by capsule endoscopy in 47% of patients with chronic renal failure [9, 10].

Although patients with OGIB can be diagnosed and treated with SBE, the rebleeding rate can reach around 40–50% following this procedure [11]. In a previous study, administration of any antiplatelet, anticoagulant, or combination therapy was not a risk factor for rebleeding [12]. In contrast, vascular lesions, CKD, and previous overt bleeding were significantly associated with rebleeding following univariate analysis and were identified as independent risk-factors of rebleeding following multivariate analysis [12]. The purpose of this study was to identify the clinical characteristics of uremic patients receiving SBE for OGIB and analyze the risk factors for rebleeding in these patients after receiving SBE.

## Materials and methods

A total of 765 patients with OGIB who underwent 1004 SBE procedures in four teaching hospitals (MacKay Memorial Hospital, Far Eastern Memorial Hospital, E-DA Hospital, and Tri-Service General Hospital, Taiwan) from 2010 to 2017 were enrolled. All SBE procedures were performed by experienced endoscopists. SBE (SIF-Q260; Olympus Corp., Tokyo, Japan) was used for small bowel examination using the push-and-pull method. Among these patients, 78 uremic patients were retrospectively evaluated for eligibility (oral approach SBE, 50 patients; anal approach SBE, 11 patients; bilateral approach SBE, 14 patients; and intraoperative enteroscopy, three patients). Data about clinical characteristics concerning comorbidities (cardiovascular disease, diabetes mellitus, hypertension, cirrhosis, and valvular heart disease (VHD)), history of antithrombotic therapy, endoscopic findings, endoscopic intervention type, location of bleeding, duration of rebleeding, and SBE-related complications were collected from the 78 uremic patients. Rebleeding was defined as the evidence of recurrent overt gastrointestinal bleeding (melena or hematochezia) with recent negative EGD and colonoscopy and/or a drop in the hemoglobin level by more than 2 g/dl. All the patients signed informed consent before SBE. The need for patient consent in this study was waived because patient information was anonymized and deidentified prior to analysis. The present study was conducted in accordance with the Declaration of Helsinki, and the institutional review boards of MacKay Memorial Hospital, Far Eastern Memorial Hospital, E-DA Hospital, and Tri-Service General Hospital approved this study (FEMH-106023-E, 17MMHIS029, EMRP-104-081, TSGHIRB No.: A202005031). The clinical characteristics of the patients were analyzed. All data were presented as the mean ± standard deviation (SD) for continuous variables or as the number (percentage) for categorical variables. Statistical analyses were performed using IBM SPSS Statistics version 22.0 (IBM Corp. Armonk, NY, USA). Continuous variables were compared using the Mann-Whitney U test and categorical variables were compared using chi-square and Fisher's exact tests. Multivariate factors were analyzed using multivariate logistic regression. We defined a P-value less than 0.05 as statistically significant, and all statistical tests were two-tailed.

## Results

We identified 765 patients who underwent 1004 SBE procedures for suspected small bowel bleeding during the study period. Of these, 78 uremic patients (who underwent 78 SBE procedures) with OGIB were enrolled. The study population selection and key findings are shown in Fig 1. The clinical characteristics of the included patients are displayed in Table 1. Seventy-two of the uremic patients were treated via hemodialysis (92.3%), and six patients were treated via peritoneal dialysis (7.7%); 21 patients were male, 57 patients were female, and the mean age was 69.2±11.6 years. Sixty-four patients underwent unilateral SBE and the remaining 14 patients underwent bilateral SBE. The overall diagnostic yield was 75.6%; the most common site of bleeder was the jejunum (50.8%). Endoscopic therapeutic procedures were performed in 49 (62.8%) patients.

Among the 78 uremic patients who underwent SBE, perforation occurred in one patient. Rebleeding events occurred in 22 patients (29.5%) among the remaining 77 patients. The most common etiologies of OGIB were angiodysplasia (74.6%), followed by diverticula (10.2%), ulcers (6.8%), other causes (5.1%), and tumors (3.4%) (Table 2). The most common bleeding site was the jejunum (50.8%), followed by the ileum (18.6%), duodenum (15.3%), colon (8.5%), and stomach (6.8%) (Table 2). The most common endoscopic intervention was argon plasma coagulation (75.6%), followed by diluted epinephrine injection (12.2%) and hemoclipping (6.1%) (Table 3). In addition, three patients underwent segmental bowel resection (two

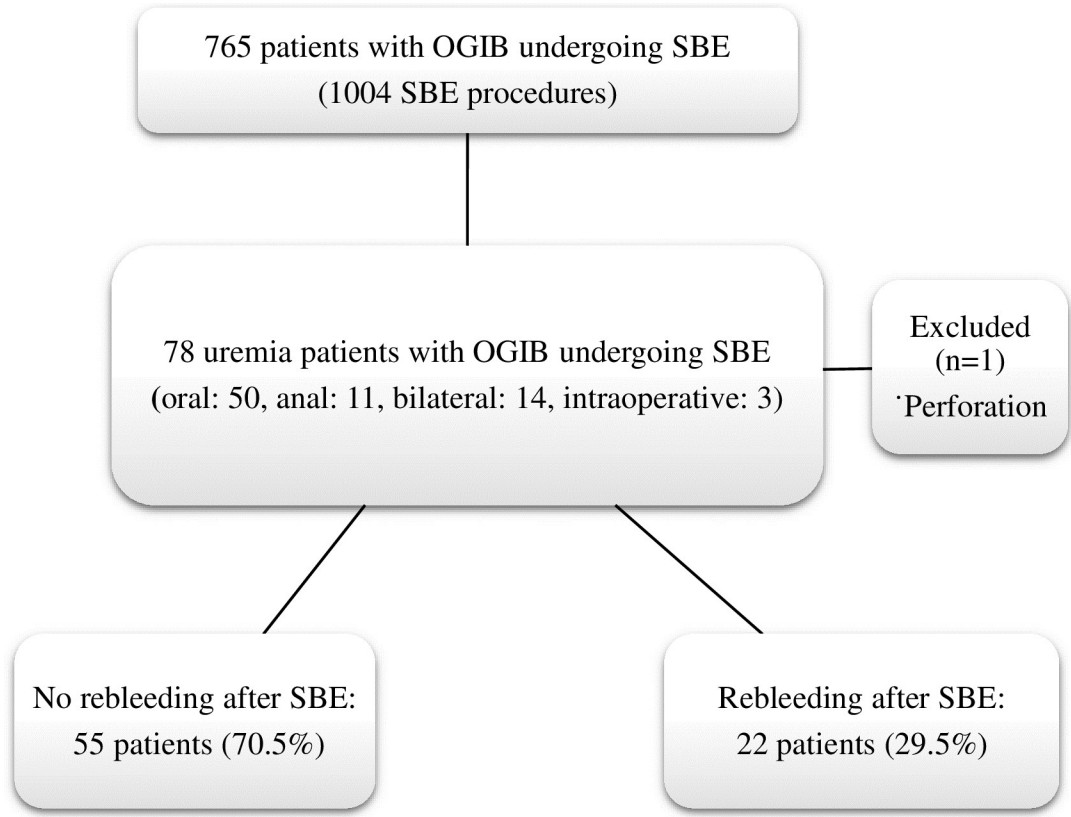

**Fig 1. A flowchart depicting the selection of the study population, clinical characteristics of the patients, and rebleeding outcomes.** OGIB: Obscure gastrointestinal bleeding; SBE: Single-balloon enteroscopy.

**Table 1. Clinical characteristics of uremic patients with obscure gastrointestinal bleeding who underwent single-balloon enteroscopy.**

| 78 patients (mean age: 69.2± 11.6 years) | |
|---|---|
| **Clinical characteristics** | **N (%)** |
| Sex (M/F) | |
| Male | 21 (26.9%) |
| Female | 57 (73.1%) |
| Method of dialysis | |
| HD | 72 (92.3%) |
| PD | 6 (7.7%) |
| Comorbidities | |
| CAD | 24 (30.8%) |
| DM | 48 (61.5%) |
| HCVD | 53 (67.8%) |
| Cirrhosis | 10 (12.8%) |
| VHD | 8 (10.3%) |
| Anti-thrombotic agents | |
| Anti-platelet (aspirin) | 16 (20.5%) |
| Anti-coagulant (clopidogrel) | 9 (11.5%) |

CAD: coronary artery disease; DM: diabetes mellitus; HCVD: hypertensive cardiovascular disease; HD: hemodialysis; PD: peritoneal dialysis; VHD: valvular heart disease

**Table 2. Endoscopic findings of uremic patients who underwent single-balloon enteroscopy procedures.**

| Endoscopic findings | N (%) |
|---|---|
| Diagnostic yield | 59 (75.6%) |
| Location of bleeding | |
| Stomach | 4 (6.8%) |
| Duodenum | 9 (15.3%) |
| Jejunum | 30 (50.8%) |
| Ileum | 11 (18.6%) |
| Colon | 5 (8.5%) |
| Cause of bleeding | |
| Angiodysplasia | 44 (74.6%) |
| Ulcer | 4 (6.8%) |
| Tumor | 2 (3.4%) |
| Diverticulum | 6 (10.2%) |
| Other | 3 (5.1%) |

cases of jejunual angiodysplasia and one of ileal varices), and one patient with jejunal multiple angiodysplasia underwent transarterial embolization after failure of endoscopic intervention.

In contrast to other variables such as age, sex, and other comorbidities (coronary artery disease, diabetes mellitus, hypertensive cardiovascular disease, and cirrhosis), where no significant differences were observed, uremic patients with VHD had a significantly higher rebleeding rate ($p < 0.05$) than those without VHD (Table 4). On multivariate analysis, VHD was a significant factor in determining the risk of rebleeding (P = 0.047) after adjusting for use of antithrombotic agents. Among the eight patients with VHD, four (50%) patients had severe aortic stenosis (AS) and the remaining patients had non-AS VHD.

The rebleeding rate in patients with OGIB, especially in patients with small bowel angiodysplasia, was found to be high despite endoscopic intervention [11]. We further determined the rebleeding rate between the angiodysplasia group and the non-angiodysplasia group in patients with endoscopic detectable bleeders. In this comparison, uremic patients with angiodysplasia had a significantly higher rebleeding rate compared to uremic patients without angiodysplasia ($p < 0.05$) (Table 5).

**Table 3. Endoscopic intervention for obscure gastrointestinal bleeding following single-balloon enteroscopy in uremic patients.**

| Treatment of bleeding | | N (%) |
|---|---|---|
| Endoscopic intervention rate | | 49 (62.8%) |
| Type of endoscopic intervention | | |
| | APC | 37 (75.6%) |
| | Hemoclipping | 3 (6.1%) |
| Diluted epinephrine injection | | 6 (12.2%) |
| | Other | 3 (6.1%) |
| Surgery or TAE* | | 4 (8.2%) |

*Three patients underwent segmental bowel resection and one patient underwent TAE

APC: argon plasma coagulation; OGIB: obscure gastrointestinal bleeding; TAE: transcatheter arterial embolization

**Table 4. Risk factors for rebleeding in uremic patients with obscure gastrointestinal bleeding.**

| | No rebleeding | Rebleeding | Univariate | | Multivariate | |
|---|---|---|---|---|---|---|
| | (n = 55) | (n = 22) | OR (95% CI) | p-value | OR (95% CI) | p-value |
| Age, mean (years) (SD) | 68.5±12.0 | 70.4±10.9 | 1.015 (0.972–1.059) | 0.508 | - | - |
| Male, N (%) | 14 (25.5) | 7 (31.8) | 1.367 (0.463–4.037) | 0.571 | - | - |
| Hemodialysis, N (%) | 50 (90.9) | 21 (95.5) | 2.1 (0.231–19.08) | 0.501 | - | - |
| CAD, N (%) | 18 (32.7) | 6 (27.3) | 0.771 (0.258–2.303) | 0.641 | - | - |
| DM, N (%) | 31 (56.4) | 16 (72.7) | 2.065 (0.702–6.073) | 0.183 | - | - |
| HCVD, N (%) | 36 (65.5) | 16 (72.7) | 1.407 (0.473–4.188) | 0.538 | - | - |
| Cirrhosis, N (%) | 8 (14.5) | 2 (9.1) | 0.588 (0.114–3.015) | 0.520 | - | - |
| VHD, N (%) | 3 (5.5) | 5 (22.7) | **5.098 (1.101–23.603)** | **0.025** | **4.775 (1.018–22.401)** | **0.047** |
| Anti-thrombotic agents, N (%) | 16 (29.1) | 9 (40.9) | 1.687 (0.602–4.727) | 0.317 | 1.5 (0.514–4.379) | 0.459 |
| Positive diagnosis, N (%) | 42 (76.4) | 16 (72.7) | 0.825 (0.268–2.544) | 0.738 | - | - |
| Endoscopic intervention, N (%) | 34 (61.8) | 15 (68.2) | 1.324 (0.464–3.779) | 0.600 | - | - |
| Location of bleeding, N (%)* | | | | | - | - |
| Stomach | 4 (9.5) | 0 (0) | 0.259 (-6.258–0.976) | 0.298 | - | - |
| Duodenum | 7 (16.7) | 2 (12.5) | 0.714 (0.132–3.868) | 0.695 | - | - |
| Jejunum | 19 (45.2) | 10 (62.5) | 2.018 (0.62–6.569) | 0.240 | | |
| Ileum | 8 (19) | 3 (18.8) | 0.981 (0.225–4.278) | 0.979 | - | - |
| Colon | 4 (9.5) | 1 (6.3) | 0.633 (0.065–6.139) | 0.691 | - | - |

* Positive in endoscopic diagnosis, total n = 58.

CAD: coronary artery disease; DM: diabetes mellitus; HCVD: hypertensive cardiovascular disease; HD: hemodialysis; PD: peritoneal dialysis; VHD: valvular heart disease

Significant values in bold text.

## Discussion

In this study, the rate of rebleeding after endoscopic treatment was high in uremic patients with OGIB (29.5%), and the most common location was the jejunum (proximal small intestine) (50.8%). This result is consistent with previous research, which concluded that even prior

**Table 5. Comparison of the clinical characteristics of patients with and without angiodysplasia.**

| | Angiodysplasia | No angiodysplasia | Univariate | | Multivariate | |
|---|---|---|---|---|---|---|
| | (n = 44) | (n = 15) | OR (95% CI) | p-value | OR (95% CI) | p-value |
| Age, mean (years) (SD) | 72±10 | 68.1±14 | 1.033 (0.977–1.092) | 0.249 | - | - |
| Male, N (%) | 8 (18.2) | 6 (40) | 0.333 (0.092–1.206) | 0.086 | | |
| Hemodialysis, N (%) | 41 (93.2) | 13 (86.7) | 2.103 (0.316–13.985) | 0.434 | - | - |
| CAD, N (%) | 15 (34.1) | 3 (20) | 2.069 (0.505–8.478) | 0.306 | - | - |
| DM, N (%) | 25 (56.8) | 9 (60) | 0.877 (0.266–2.892) | 0.829 | - | - |
| HCVD, N (%) | 33 (75) | 9 (60) | 2 (0.58–6.898) | 0.268 | - | - |
| Cirrhosis, N (%) | 5 (11.4) | 1 (6.7) | 1.795 (0.193–16.729) | 0.603 | - | - |
| VHD, N (%) | 8 (18.2) | 0 (0) | 7.219 (-0.212–6.862) | 0.084 | | |
| Rebleeding, N (%) | 16 (36.4) | 1 (6.7) | **8 (0.961–66.629)** | **0.028** | **8.82 (1.014–76.727)** | **0.049** |

CAD: coronary artery disease; DM: diabetes mellitus; HCVD: hypertensive cardiovascular disease; HD: hemodialysis; PD: peritoneal dialysis; SBE: single balloon enteroscopy; VHD: valvular heart disease

Significant values in bold text

to the advances in small bowel evaluation, patients with CKD were found to have a higher incidence of angiodysplasia of the stomach or proximal small bowel [6].

Bleeders located in the stomach, proximal duodenum, or colon could be diagnosed and treated by traditional EGD or colonoscopy; however, the management of distal small bowel bleeders is different. Small bowel lesions could be diagnosed and treated by enteroscopy specialists who have greater experience and skills. Also, the identification of small bowel lesions relies on the physician's experience and expertise. In clinical practice, small pale angiodysplasias without active bleeding in the small bowel, which are easily confused as erosions or erythematous mucosal injury, are frequently ignored. Only typical bright red spots or lesions with active bleeding can easily be diagnosed and treated. The identification of small intestinal lesions, especially small vascular lesions, is highly dependent on the expertise of the clinical physician [13].

In addition, since angiodysplasia lesions are sometimes located in multiple areas, these are easily ignored by unfamiliar operators or in cases of incomplete examination of the whole small bowel. Correspondingly, a retrospective analysis concluded that patients with non-isolated small bowel gastrointestinal angiodysplasias had four times the odds of rebleeding within 1 year following capsule endoscopy compared to those with isolated angiodysplasias [14].

Previous studies have revealed that vascular lesions, such as angiodysplasia, are a major small bowel enteroscopic finding in patients with OGIB [15]. The detailed mechanism underlying angiodysplasia requires further research, but several studies have reported that certain angiogenic factors, including angiopoietin 1 and 2, may be implicated in its pathophysiology [16]. Angiopoietin-2 (Ang2) rises rapidly in response to angiogenic stimuli and is believed to induce the formation of immature and unstable blood vessels. This may be the source of easy rebleeding in the angiodysplasia groups. Ang2 is related to endothelial physiology and cardiovascular remodeling. Accordingly, endothelial dysfunction is associated with various cardiovascular risk factors. An increase in Ang2 can be observed in most cardiovascular diseases, such as coronary heart disease, congestive heart failure, and peripheral arterial disease, and associated conditions, such as CKD [17, 18].

Impaired function of platelet glycoproteins GPIIb/IIIa in uremic patients alters the release of adenosine diphosphate and serotonin and impairs the metabolism of prostaglandin and arachidonic acid, resulting in impaired platelet adhesion and aggregation. At the same time, progressive renal impairment increases platelet dysfunction. Therefore, OGIB is an expected complication in CKD patients compared to the general population [19]. Other studies have shown that CKD is a positive predictor of etiology in patients with small bowel bleeding. Previous studies have also shown that there is an increased prevalence of small bowel angiodysplasia in CKD, which may be due to increased Ang2 in CKD patients, as described above [10]. It remains unclear whether the two risk factors (CKD and small bowel angiodysplasia) have an additive effect that causes a higher rebleeding rate in patients with CKD and angiodysplasia. Consequently, additional studies are required in the future.

In this study, we found that VHD resulted in a higher rebleeding rate than other clinical variants. Among the eight patients with VHD, four (50%) were patients with severe AS. The result was consistent with a previous study that found an association between AS and angiodysplasia. Gastrointestinal bleeding from angiodysplasias in patients with AS is referred to as 'Heyde's syndrome.' Heyde described AS and gastrointestinal bleeding in the year 1958 [20]. As a possible mechanism for this association, shear stress resulting from turbulence caused by aortic valvular disease prompts proteolysis of high-molecular-weight multimers of von Willebrand factor (VWF), which leads to impaired hemostasis. This phenomenon reflects an increased risk of bleeding from VWF dysfunction in similar high-shear stress states, such as hypertrophic obstructive cardiomyopathy, ventricular septal defects, and para-valvular leaks.

Any cardiovascular disease that accelerates VWF clearance may lead to bleeding from coexisting gastrointestinal angiodysplasia [21]. A previous study found 42 patients with severe aortic stenosis and concluded that VWF abnormalities are directly related to the severity of aortic stenosis [22]. In contrast, the probability of mitral valve disease without shear stress does not increase in bleeding intestinal angiodysplasia. Given that the main cause of Heyde's syndrome is the lack of high molecular weight polymorphs of VWF due to high shear stress, the most effective treatment is the correction of AS, i.e., surgical aortic valve replacement or transcatheter aortic valve implantation instead of correction of coagulopathy with blood transfusion [23]. Since only eight patients had VHD, we did not further determine the relationship between rebleeding and the different types of VHD. We adjusted for anti-thrombotic agent use in patients with VHD using multivariate logistical regression analysis since it can be a confounding factor. VHD still showed a significant difference between the rebleeding and non-rebleeding groups (p = 0.047).

Due to the thin wall of the small intestine, the endoscopist must carefully consider the possibility of complications or perforation while performing diagnostic or therapeutic enteroscopy. Argon plasma coagulation (APC) uses a jet of ionized argon gas guided through a probe, which passes through the endoscope, allowing the gas to be transmitted to the target lesion without direct contact with the mucosa. The depth of coagulation is limited to the superficial mucosa, and coagulation can be controlled using power settings, gas flow, and duration. Due to the low incidence of complications, APC has become the most widely used method for the endoscopic therapy of angiodysplasia lesions in the small intestine [24]. This is reflected in the current study, where most of the patients received APC to treat angiodysplasia bleeding (75.6%).

As per the Yano-Yamamoto classification, angiodysplasia was classified as type I (type Ia and Ib). Dieurofoy's lesion was classified as type 2, AVM was classed as type 3, and unclassified lesion was classed as type 4. Therefore, our enrolled patients with angiodysplasia had type 1 vascular lesions. We did not determine the effect of different subtypes of angiodysplasia (type Ia and Ib) on rebleeding. However, whether the different types of vascular lesions in the small bowel can affect the rebleeding rate in uremic patients with OGIB needs further determination.

In summary, this retrospective study found that angiodysplasia is the most common cause of OGIB in patients with uremia undergoing SBE. In this study, the lesions were mainly found in the jejunum, which is consistent with the results of previous studies. We also found that VHD might be an independent risk factor associated with rebleeding after SBE in uremic patients experiencing OGIB. There may be a correlation between AS and the rebleeding rate of angiodysplasia, but because of the limited database of this study, more data is needed to prove this association. The rebleeding rate of uremic patients with angiodysplasia bleeding was significantly higher than that of patients without angiodysplasia bleeding; however, further research is required to confirm these results. APC is currently the most widely used method for treating lesions in the small intestine because of its high safety profile.

## Supporting information

**S1 Data.**
(XLSX)

## Acknowledgments

We are grateful to our assistant, Yi-Jin Chen for her assistance in collecting the clinical data of enrolled patients.

## Author Contributions

**Conceptualization:** Chen-Shuan Chung, Chi-Ming Tai, Chen-Wang Chang, Chao-Ming Tseng, Meng-Chiung Lin.

**Formal analysis:** Tien-Yu Huang.

**Investigation:** Tien-Yu Huang.

**Methodology:** Tien-Yu Huang.

**Resources:** Chen-Shuan Chung, Chi-Ming Tai, Chen-Wang Chang, Chao-Ming Tseng, Go-Shine Huang, Meng-Chiung Lin.

**Supervision:** Tien-Yu Huang.

**Writing – original draft:** Hsuan-Jen Hung.

**Writing – review & editing:** Tien-Yu Huang.

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
