## [Decision Letter · Decision Letter 0]

19 Jul 2022

PONE-D-22-16076Clinical characteristics and risk factors for rebleeding in uremic patients with obscure gastrointestinal bleeding undergoing deep enteroscopy: a multi-center retrospective study in TaiwanPLOS ONE

Dear Dr. Huang,

Thank you for submitting your manuscript to PLOS ONE. After careful consideration, we feel that it has merit but does not fully meet PLOS ONE’s publication criteria as it currently stands. Therefore, we invite you to submit a revised version of the manuscript that addresses the points raised during the review process.

The manuscript has been evaluated by two experts in the filed and significant improvement is required. 

We look forward to receiving your revised manuscript.

Kind regards,

Hsu-Heng Yen

Academic Editor

PLOS ONE

Journal Requirements:

2. In the ethics statement in the manuscript, you have stated that patient consent was not required as the ethics committee waived the need for consent. Please clarify whether all four of the ethics committees waived participant consent for your study and whether patient records from all four databases used in your study were anonymised and de-identified before access.

A clean copy of the edited manuscript (uploaded as the new *manuscript* file).

The research funding for this work was supported by Tri-Service General Hospital, Taiwan (TSGH-D-109073) and the Department of Defense, Taiwan (MND-MAB-D-111114).

 The authors received no specific financial funding for this work.

Reviewers' comments:

Reviewer's Responses to Questions

**Comments to the Author**

1. Is the manuscript technically sound, and do the data support the conclusions?

Reviewer #1: No

Reviewer #2: Partly

2. Has the statistical analysis been performed appropriately and rigorously? 

Reviewer #1: No

Reviewer #2: I Don't Know

3. Have the authors made all data underlying the findings in their manuscript fully available?

Reviewer #1: Yes

Reviewer #2: Yes

4. Is the manuscript presented in an intelligible fashion and written in standard English?

Reviewer #1: Yes

Reviewer #2: No

5. Review Comments to the Author

Reviewer #1: Authors proposed clinical characteristics and risk factors for rebleeding in uremic patients with obscure gastrointestinal bleeding undergoing deep enteroscopy. It would give another point of view for clinical daily practice. I have several questions as below.

1. In statistical analysis, continuous variables should be analyzed using the Mann–Whitney U test. Therefore, I suggest authors may recheck the P value for continuous variables in table 4 and 5.

2. In table 4 and 5, authors presented the characteristics and "factors" between subgroups with/without rebleeding or with/without angiodysplsia. Moreover, authors should calculate the odds ratios for these risk factors which were determined through univariate and multivariate logistic regression analyses in another tables.

3. In table 4, I don't know the meaning (no data) about "location of bleeding" between subgroups with/without rebleeding.

Reviewer #2: Appreciate the authors providing us about the risk factors for rebleeding in uremic patients with obscure GI bleeding

My comment:

1. There is no clear mention about definition of "rebleeding" in the article. There is no detailed statements about lesions with rebleeding. What is the most common lesion type of rebleeding, angiodysplasia or ulcer?

2. Eight patients with VHD and 4 of then with severe aortic stenosis and 4 patient were non-AS patients. Have you performed subgroup analysis for the relationship between OGIB and non-AS patients ? Many patients with VHD take antiplatelet or anticoagulation for prevention of thrombotic events. Have you analyze the effect of OGIB in anti-thrombotic agent for patient with VHD ? Your statistical analysis should manage the possible confounding factors of rebleeding.

3. No “Age” included in the clinical characteristics in table 1.

4. Table 1: What is the details of anti-thrombotic agents (antiplatelet ? warfarin ? DOAC ? ) Dose each subtype of anti-thrombotic agent have effect on OGIB?

5. Table 3: Endoscopic intervention for what kinds of bleeder ? Four patient received surgery or TAE and what is the cause of bleeding ? Different etiology of bleeding may be one of the risk factor of rebleeding.

6. You did not mention the endoscopic appearance of angiodysplasia. There are different types of angiodysplasia (Yano –Yamamoto classification) and each had different rebleeding rate. You should analyze the rebleeding risk between different subtype of angiodysplasia.

7. There are multiple grammatical mistakes throughout the manuscript, I would recommend the authors revise the manuscript and consult English editing.

6. PLOS authors have the option to publish the peer review history of their article (what does this mean?). If published, this will include your full peer review and any attached files.

Reviewer #1: No

Reviewer #2: No

---

## [Author Response · Author response to Decision Letter 0]

20 Sep 2022

Responses to Reviewers’ comments:

Reviewer #1: Authors proposed clinical characteristics and risk factors for rebleeding in uremic patients with obscure gastrointestinal bleeding undergoing deep enteroscopy. It would give another point of view for clinical daily practice. I have several questions as below.

1. In statistical analysis, continuous variables should be analyzed using the Mann–Whitney U test. Therefore, I suggest authors may recheck the P value for continuous variables in table 4 and 5.

A:

Thank you for your reminder. We rechecked the statistical analysis in detail and revised the manuscript in the “Materials and Methods” section and in Tables 4 and 5.

2. In table 4 and 5, authors presented the characteristics and "factors" between subgroups with/without rebleeding or with/without angiodysplsia. Moreover, authors should calculate the odds ratios for these risk factors which were determined through univariate and multivariate logistic regression analyses in another tables.

A:

Thank you for your comments. We reperformed the statistical analysis and added the odds ratios from univariate and multivariate analyses in Tables 4 and 5. In Table 4, two factors (valvular heart disease and use of anti-thrombotic agents—correlated with VHD in clinical use) were entered into the multivariate logistical regression model. In Table 5, only the P-value of “rebleeding” was significant on univariate analysis, which was then entered into the multivariate logistical regression model.

3. In table 4, I don't know the meaning (no data) about "location of bleeding" between subgroups with/without rebleeding.

A:

Thank you for your question. The location of bleeding refers to the location of bleeder. We revised Table 4 and performed statistical analysis further in different locations.

Reviewer #2: Appreciate the authors providing us about the risk factors for rebleeding in uremic patients with obscure GI bleeding

1. There is no clear mention about definition of "rebleeding" in the article. There is no detailed statements about lesions with rebleeding. What is the most common lesion type of rebleeding, angiodysplasia or ulcer?

A:

Thanks for your comment. The definition of rebleeding was described in “Materials and Methods” in the revised manuscript. However, not all patients with rebleeding underwent repeat deep enteroscopy; the actual etiology of rebleeding could thus not be clarified clearly. However, referring to the past literature and our indirect data (in the patients with different bleeding types, the rebleeding rate of angiodysplasia type vs. non-angiodysplasia type was 34% vs. 6.7%), the most common type of re-bleeding was likely to have been angiodysplasia. We put this issue into the “Discussion” section.

2. Eight patients with VHD and 4 of then with severe aortic stenosis and 4 patient were non-AS patients. Have you performed subgroup analysis for the relationship between OGIB and non-AS patients ? Many patients with VHD take antiplatelet or anticoagulation for prevention of thrombotic events. Have you analyze the effect of OGIB in anti-thrombotic agent for patient with VHD ? Your statistical analysis should manage the possible confounding factors of rebleeding.

A: 

Given that only four patients had non-AS VHD, we did not further determine the relationship between OGIB and non-AS patients. As for the confounding effect of anti-thrombotic agents in patients with VHD, we adjusted for this factor using multivariate logistical regression analysis, and VHD still showed a statistically significant difference between the rebleeding and non-rebleeding groups (p=0.047). 

3. No “Age” included in the clinical characteristics in table 1.

A: Thank you for pointing this out. The age has been included in the top row of Table 1 (marked in Red). 

4. Table 1: What is the details of anti-thrombotic agents (antiplatelet? warfarin? DOAC?) Dose each subtype of anti-thrombotic agent have effect on OGIB?

A: In 25 patients who received antithrombotic agents (32.1%), 16 received aspirin and 9 received clopidogrel. We revised Table 1 accordingly. In our statistical analysis, the diffent types of anti-thrombotic agents showed no significalnt differences between the rebleeding and no rebleeding-groups (data no shown). 

5. Table 3: Endoscopic intervention for what kinds of bleeder? Four patient received surgery or TAE and what is the cause of bleeding? Different etiology of bleeding may be one of the risk factor of rebleeding.

A:

Thank you for your comments. In our study, three patients underwent segmental bowel resection (two cases of jejunual angiodysplasia and one of ileal varices), and one patient with jejunal multiple angiodysplasia underwent transarterial embolization after failure of endoscopic intervention. This point was added to the “Results” section of the revised manuscript.

6. You did not mention the endoscopic appearance of angiodysplasia. There are different types of angiodysplasia (Yano –Yamamoto classification) and each had different rebleeding rate. You should analyze the rebleeding risk between different subtype of angiodysplasia.

A:

Thank you for your constructive comment. As per the Yano-Yamamoto classification, angiodysplasia was classified as type I (type Ia and Ib). Dieurofoy’s lesion was classified as type 2, AVM was classed as type 3, and unclassified lesion was classed as type 4. Therefore, our enrolled patients with angiodysplasia had type 1 vascular lesion and we did not determine the effect of different subtypes of angiodysplasia (type Ia and Ib) on rebleeding. However, whether the different types of vascular lesions in the small bowel can affect the rebleeding rate in uremic patients with OGIB needs further determination. We discussed this concern in the “Discussion” section.

7. There are multiple grammatical mistakes throughout the manuscript, I would recommend the authors revise the manuscript and consult English editing.

A:

Thanks for your positive comment. We have gotten the manuscript edited by a professional English language-editing service.

---

## [Decision Letter · Decision Letter 1]

17 Oct 2022

PONE-D-22-16076R1Clinical characteristics and risk factors for rebleeding in uremic patients with obscure gastrointestinal bleeding undergoing deep enteroscopy: a multi-center retrospective study in TaiwanPLOS ONE

Dear Dr. Huang,

Thank you for submitting your manuscript to PLOS ONE. After careful consideration, we feel that it has merit but does not fully meet PLOS ONE’s publication criteria as it currently stands. Therefore, we invite you to submit a revised version of the manuscript that addresses the points raised during the review process.

We look forward to receiving your revised manuscript.

Kind regards,

Hsu-Heng Yen

Academic Editor

PLOS ONE

Journal Requirements:

Reviewers' comments:

Reviewer's Responses to Questions

**Comments to the Author**

1. If the authors have adequately addressed your comments raised in a previous round of review and you feel that this manuscript is now acceptable for publication, you may indicate that here to bypass the “Comments to the Author” section, enter your conflict of interest statement in the “Confidential to Editor” section, and submit your "Accept" recommendation.

Reviewer #2: (No Response)

Reviewer #3: All comments have been addressed

2. Is the manuscript technically sound, and do the data support the conclusions?

Reviewer #2: Partly

Reviewer #3: Yes

3. Has the statistical analysis been performed appropriately and rigorously? 

Reviewer #2: I Don't Know

Reviewer #3: Yes

4. Have the authors made all data underlying the findings in their manuscript fully available?

Reviewer #2: Yes

Reviewer #3: Yes

5. Is the manuscript presented in an intelligible fashion and written in standard English?

Reviewer #2: No

Reviewer #3: Yes

6. Review Comments to the Author

Reviewer #2: Appreciate the authors revising the manuscripts with efforts.

#1 Page 7 line 92-95

The sentence can be deleted “After being informed of the needs and potential complications of SBE, all patients signed an informed consent form. The requirement for patient consent for publication was waived because the patient information was anonymized and de-identified prior to analysis.”. You have repeated the same words in page 7, line 106-108.

# 2 Page 7 line 104

Why the definition of rebleeding is “ the evidence of recurrent overt gastrointestinal bleeding with recent negative EGD and colonoscopy and/or a drop in the Hb level…..” ? Why not enteroscopy (SBE)?

The purpose of manuscript as you mentioned was to “identify the clinical characteristics of uremic patients receiving SBE for OGIB and analyze the risk factors “

#3 Figure 1: I suggest that the author try to simplify the figure and remove redundant words. Otherwise this picture looks unnecessary.

#4 Page 17 line 194-196 ??

“As lesions in the small intestine are similar to those in other gastrointestinal areas such as the stomach or large intestine, they can be diagnosed and treated by traditional EGD or colonoscopy. “

Does the author really express what you want to express correctly?

#5 There are multiple grammatical mistakes throughout the manuscript, I strongly recommend the author revise the manuscript and consult English editing again.

Reviewer #3: The authors purpose to identify the clinical characteristics of uremic patients undergoing deep enteroscopy for OGIB and analyze the risk factors for rebleeding after deep enteroscopy. Authors concluded that uremic patients with angiodysplasia-related bleeding appear to have a higher rebleeding rate and VHD is an independent risk factor associated with rebleeding after deep enteroscopy in uremic patients with OGIB. I think this article might give us an alternative point of view in our clinical management in uremic patients with OGIB.

7. PLOS authors have the option to publish the peer review history of their article (what does this mean?). If published, this will include your full peer review and any attached files.

Reviewer #2: No

Reviewer #3: **Yes: **Wei-Chen Tai

---

## [Author Response · Author response to Decision Letter 1]

26 Oct 2022

Responses to Reviewers’ comments:

Reviewer #2: Appreciate the authors revising the manuscripts with efforts.

#1 Page 7 line 92-95

The sentence can be deleted “After being informed of the needs and potential complications of SBE, all patients signed an informed consent form. The requirement for patient consent for publication was waived because the patient information was anonymized and de-identified prior to analysis.”. You have repeated the same words in page 7, line 106-108.

A:

Thank you for your comment. We have revised the manuscript.

# 2 Page 7 line 104

Why the definition of rebleeding is “ the evidence of recurrent overt gastrointestinal bleeding with recent negative EGD and colonoscopy and/or a drop in the Hb level…..” ? Why not enteroscopy (SBE)?

The purpose of manuscript as you mentioned was to “identify the clinical characteristics of uremic patients receiving SBE for OGIB and analyze the risk factors “

A:

Thank you for your comment. Among the 78 uremic patients with OGIB receiving SBE, not all patients with recurrent GI bleeding underwent repeat EGD, colonoscopy, or SBE. Therefore, we defined rebleeding as the evidence of recurrent overt gastrointestinal bleeding (melena or hematochezia) with recent negative EGD and colonoscopy and/or a drop in the hemoglobin level by >2 g/dl. 

#3 Figure 1: I suggest that the author try to simplify the figure and remove redundant words. Otherwise this picture looks unnecessary.

A:

As suggested, we have simplified Figure 1 in the revised manuscript. 

#4 Page 17 line 194-196 ??

“As lesions in the small intestine are similar to those in other gastrointestinal areas such as the stomach or large intestine, they can be diagnosed and treated by traditional EGD or colonoscopy. “

Does the author really express what you want to express correctly?

A: 

Thank you for your kind comment. We have revised the content of this paragraph to express our meaning correctly.

#5 There are multiple grammatical mistakes throughout the manuscript, I strongly recommend the author revise the manuscript and consult English editing again.

A:

Thank you for pointing it out. I have further consulted an English editing service and revised this manuscript.

---

## [Editor Report · Decision Letter 2]

27 Oct 2022

Clinical characteristics and risk factors for rebleeding in uremic patients with obscure gastrointestinal bleeding undergoing deep enteroscopy: a multi-center retrospective study in Taiwan

PONE-D-22-16076R2

Dear Dr. Huang,

We’re pleased to inform you that your manuscript has been judged scientifically suitable for publication and will be formally accepted for publication once it meets all outstanding technical requirements.

Kind regards,

Hsu-Heng Yen

Academic Editor

PLOS ONE
---

## [Editor Report · Acceptance letter]

18 Nov 2022

PONE-D-22-16076R2 

Clinical characteristics and risk factors for rebleeding in uremic patients with obscure gastrointestinal bleeding undergoing deep enteroscopy: a multi-center retrospective study in Taiwan 

Dear Dr. Huang:

I'm pleased to inform you that your manuscript has been deemed suitable for publication in PLOS ONE. Congratulations! Your manuscript is now with our production department. 

Kind regards, 

on behalf of

Dr. Hsu-Heng Yen 

Academic Editor

PLOS ONE